# Effects of Astaxanthin from Shrimp Shell on Oxidative Stress and Behavior in Animal Model of Alzheimer’s Disease

**DOI:** 10.3390/md17110628

**Published:** 2019-11-04

**Authors:** Takunrat Taksima, Pennapa Chonpathompikunlert, Morakot Sroyraya, Pilaiwanwadee Hutamekalin, Maruj Limpawattana, Wanwimol Klaypradit

**Affiliations:** 1Department of Fishery Products, Faculty of Fisheries, Kasetsart University, Bangkok 10900, Thailand; takunrat@hotmail.com; 2Expert Centre of Innovative Health Food (InnoFood) Thailand Institute of Scientific and Technological Research (TISTR) 35 Moo 3 Technopolis, Khlong 5, Khlong Luang, Pathum Thani 12120, Thailand; 3Department of Anatomy, Faculty of Science, Mahidol University, Bangkok 10400, Thailand; morakot.sry@mahidol.ac.th; 4Department of Physiology, Faculty of Science, Prince of Songkla University, Hat Yai, Songkhla 90112, Thailand; pilaiwanwadee.h@psu.ac.th; 5Department of Food Technology, Faculty of Science, Siam University, Bangkok 10160, Thailand; maruj@siam.edu; 6Center for Advanced Studies for Agriculture and Food (CASAF), Kasetsart University Institute for Advanced Studies, Kasetsart University, Bangkok 10900, Thailand

**Keywords:** astaxanthin, Alzheimer’s disease, amyloid-β (1-42) peptides, encapsulation, shrimp shells

## Abstract

This study aimed to investigate the effect of astaxanthin (ASX) extracted and ASX powder from shrimp (*Litopenaeus vannamei*) shells on Wistar rats with Alzheimer’s disease, induced by amyloid-β (1-42) peptides. In this task, the rats were divided into eight groups: (1) Control, (2) sham operate, (3) negative control (vehicle) + Aβ_1-42_, (4) ASX extract+Aβ_1-42_, (5) commercial ASX + Aβ_1-42_, (6) ASX powder + Aβ_1-42_, (7) blank powder + Aβ_1-42_, and (8) vitamin E + Aβ_1-42_. All treatments were orally administrated for 30 days. At 14- and 29-days post injection, animals were observed in behavioral tests. On the 31st day, animals were sacrificed; the hippocampus and cortex were collected. Those two brain areas were then homogenized and stored for biochemical and histological analysis. The results showed that the Aβ_1-42_ infused group significantly reduced cognitive ability and increased memory loss, as assessed by the Morris water maze test, novel object recognition test, and novel object location test. Moreover, the Aβ_1-42_ infused group exhibited a deterioration of oxidative markers, including glutathione peroxidase enzymes (GPx), lipid peroxidation (MDA), products of protein oxidation, and superoxide anion in the cortex and the hippocampus. Meanwhile, ASX powder (10 mg/kg body weight) showed a significant reduction in cognitive and memory impairments and oxidative stress which is greater than ASX extract in the same dose of compound or vitamin E (100 mg/kg body weight). Our study indicates the beneficial properties of ASX in alleviation of cognitive functions and reducing neurodegeneration in Wistar rats induced by amyloid-β (1-42) peptides.

## 1. Introduction

Alzheimer’s disease (AD) is a progressive neurodegenerative disease, which can damage memory and cognitive function [1] while the biological mechanisms involved in AD are not yet fully understood. However, there is evidence to support the hypothesis that free radical-induced oxidative damage may be important [2]. AD is associated with the overproduction of β-amyloid protein (Aβ), which accumulate in the hippocampus and cerebral cortex [3,4]. Aβ appear to play a role in inducing oxidative stress through the formation of reactive oxygen and nitrogen species [5]. Oxidative stress has been reported to have many negative effects in humans, including lipid peroxidation, protein oxidation, inflammation, disturbance of cell functions, apoptosis, and formation of neurofibrillary tangles. These disturbances can result in the loss of synaptic connections between neurons in the hippocampus and cerebral cortex, leading to reduction in cognitive function and dementia [1]. One reliable way to protect cells from the damages of oxidative stress is to increase the potential of the endogenous oxidative defense through antioxidants, which can be part of the diet of pharmacological supplements. During the past decade, novel applications of naturally-occurring antioxidant compounds have been shown to have potential benefits.

Astaxanthin (ASX) is a ketocarotenoid pigment, and naturally exists in plants, microorganisms, and aquatic animals such as crabs, shrimp, and salmon. ASX has been reported to exhibit potential pharmacological activity, including anti-inflammatory, antioxidant and anti-apoptotic, anti-cancer [6], anti-diabetic [7], and immunomodulation. Moreover, according to its ability to traverse the blood-brain-barrier, ASX also has neuroprotective effects [8,9,10,11]. Although previous studies demonstrated that ASX can prevent AD symptoms, all of published reports used ASX from *Haematococcus pluvialis*, which is quite difficult to cultivate in Thailand, the location of our study. However, a promising alternative source of ASX is processing waste from Pacific white shrimp (*Litopenaeus vannamei*), which is a major export for the country. Thus far, scientific support of AD protection from this source of ASX has not yet been established. 

The ASX molecule, however, is unstable during the extraction process and in storage because its nonpolar structure is highly unsaturated, resulting in easily lost biological activity and thus considerably limiting its application in foods and dietary supplements. Encapsulation technology, a method of packing sensitive compounds within wall materials to protect from adverse conditions, is therefore applied to tackle such problems. Moreover, encapsulation has further benefits of enabling controlled release and delivery to the desired site of action. Microencapsulation of ASX can be accomplished by various processes including emulsion/solvent evaporation [12], coacervation [13], liposome formation [14], solvent displacement [15], and spray drying [16]. However, little is known about the use of cryogenic incorporated freeze-drying to create a powdered form of ASX. Thus, this study aimed to investigate the effects of ASX extract and encapsulated ASX on cognitive impairment and neurodegeneration in an animal model of amyloid beta-induced AD.

## 2. Results 

### 2.1. Effects of ASX on Learning and Memory Deficit in Aβ_1-42_-Induced AD Rats 

The effect of ASX on spontaneous motor behaviors including grooming, rearing, and licking behaviors were demonstrated in Appendix A in supplementary data. The results showed that all groups failed to show significant changes in all spontaneous motor behaviors. These indicated that the cognitive results from Morris water maze test, object recognition test, and object location test could be excluded from the false positive effect.

The effects of ASX on spatial memory of rats were assessed by the escape latency and time spent in target quadrant in the Morris water maze (MWT) test as illustrated in Figure 1. Discrimination index scores recorded from the novel object location test (NOL) were shown in Figure 2C,D. The effect of ASX on non-spatial memory as determined by the object recognition test was shown in Figure 2A,B.

Figure 1A,B, show that artificial cerebrospinal fluid (ACSF) produced no significant change in either escape latency or time spent in the target quadrant in the Morris water maze test. Intracerebroventricular administration of Aβ_1-42_ significantly increased escape latency and decreased time spent in target quadrant (*p* < 0.05, Figure 1A,B). These showed that the memory impairment was induced by Aβ_1-42_. Treatments with vitamin E (VE), ASX extract (AE), commercial ASX (AC), and ASX powder (AP) significantly decreased escape latency and increased time spent in target quadrant (*p* < 0.05) for all treatments. Interestingly, the AC and AP groups had significantly lower escape latency and higher time spent in target quadrant than the AE and VE treatment groups (*p* < 0.05).

For the object recognition and location memory tests, the discrimination index was lower in Aβ_1-42_ treated rats compared with a sham control group (SO), while the groups treated with VE, AE, AC, and AP had significantly higher discrimination index. With the blank powder encapsulation, the discrimination index was not increased. In contrast, the discrimination index for the AC and AP treatments was significantly higher than the AE group for both the object recognition and location tests (Figure 2A–D). The aforementioned results confirm that treatments with ASX from white shrimp shells significantly improved learning and reduced memory dysfunction in the AD model induced by Aβ_1-42_ (*p* < 0.05). The AP treatment showed better learning and memory than AE (*p* < 0.05) and was equivalent with AC. Notably, the AE treated group did not improve the learning and memory functions than those of AP and AC treated mice.

### 2.2. Effect of Astaxanthin in Reducing Brain Oxidative Stress 

The lipid peroxidation product malondialdehyde (MDA), level of protein carbonyl, glutathaione peroxidase (GPx) assays, and percent inhibition of superoxide anion were used to evaluate the free-radical scavenging capacity of ASX in both the cortex and hippocampus to compare non-treated, V-treated, VE-treated, and blank powder (BP)-treated groups. As shown in Figure 3, Figure 4, Figure 5 and Figure 6, AE, AC, AP, and VE treated groups had decreased MDA, and protein carbonyl, while increased percent inhibition of superoxide anion and GPx activity when compared to the V-treated and BP-treated groups (*p* < 0.05). These results indicate that the effectiveness of ASX from white shrimp shell was not significantly different (*p* ≥ 0.05) from commercial ASX.

### 2.3. Effects of ASX on Neuronal Survival and Amyloidosis in Hippocampal and Cortex Regions

The CA1 and CA3 in hippocampus, as well as the cortex regions were used as the reference zones because they are involved in memory and neuronal loss in these regions resulted in Alzheimer’s disease. The low power views of the mouse hippocampus and cortex were shown in Appendix A. Cresyl violet staining was used to evaluate the surviving neurons in the cortex and hippocampal CA1, CA3. Figure 7 shows that surviving cells in the control group (a, I, and q) showed round and pale-stained nuclei in the cortex and hippocampus whereas the Aβ_1-42_ treated group showed serious cell death (c, k, and s). The SO treatment was shown as not significantly different in the surviving neurons when compared to the control group. AC and AP (10 mg/kg) treatments showed a significant decrease in neuronal degeneration in CA1 and CA3 regions of hippocampus and the cortex, whereas the AE group showed a significant decrease in neuronal degeneration in the cortex only (t). The VE group treatment showed a significant decrease in neuronal degeneration in CA1 of hippocampus and cortex (g and w). The BP treated group with alginate and modified starch vehicle did not show any protection against neuronal injury (h, p, and x). 

On the other hand, significant accumulation of positive staining of β-amyloid (% area) was observed in the hippocampal and cortex of Aβ_1-42_ peptide infused rats using immunohistochemistry staining. Hippocampal CA1, CA3, and cortex of rats treated with ASX in various forms (AE, AC, AP, and VE) for 30 days showed a reduction of positive staining of β-amyloid when compared with other Aβ_1-42_ treated groups. Interestingly, AC and AP showed better biological activity than other treatments (Figure 8).

## 3. Discussion 

In recent years, ASX has gained attention for its potential therapeutic role in neurodegenerative diseases. Numerous reports that ASX treatment is effective at protecting neurons from various forms of CNS damage in models of specific neuronal damage and neurodegenerative disease [17,18,19]. There are few studies investigating the benefits of ASX from Pacific white shrimp (*Litopenaeus vannamei*) shell in neurodegenerative disease model, while there are many using *Haematococcus pluvialis* [9,20,21]. In fact, our ASX extract from Pacific white shrimp shell processing waste contains higher proportion of ASX diester and higher PUFA content (61.74%) [22] than Arabian red shrimp (PUFA content of 29.85%) and *Haematococus pluvialis* [23]. This implies that the powerful antioxidant properties of carotenoid extract of *Litopenaeus vannamai* may be attributed to the antioxidant synergism of ASX and PUFAs contained when in compared to other sources of ASX [22,24].

In this study, we determined the effects of ASX from white shrimp shells on spatial and non-spatial memory, and on neurodegeneration in various sub-regions of brain in the animal model of AD induced by Aβ_1-42_. The results clearly demonstrated that ASX from white shrimp shell significantly improved spatial memory (Figure 1; Figure 2C,D) as indicated by Morris water maze and object recognition tests, improved non-spatial memory (Figure 2A,B) as indicated by object recognition test, and reduced neurodegeneration, as indicated by surviving neurons and amyloid beta plaque level (Figure 8).

Recently, oxidative stress has been detected in the early stage of AD through Aβ accumulation and progression via mitochondrial dysfunction, caused by the generation of reactive oxygen species (ROS) and reduction in the level of detoxifying enzymes, including superoxide dismutase (SOD), catalase (CAT), and GPx [25,26]. ROS destroys the polyunsaturated fatty acids of cellular membranes to generate lipid peroxidation products such as MDA, which may serve as an indicator of the oxidative damage level and induce neuronal deterioration [27]. In addition, protein carbonyl is another hallmark of oxidative damage [28]. As expected, the observed neuroprotection by ASX from white shrimp shell was due to its antioxidant property through support of the activity of GPx, and percent inhibition of superoxide anion, as well as reduction of MDA and protein carbonyl levels, compared to vehicle-AD groups (*p* < 0.05) as shown in Figure 3, Figure 4, Figure 5 and Figure 6. This is consistent with our previous study, where we showed that the ASX from white shrimp shells eliminates superoxide anion, which causes lipid peroxidation [29]. It is also consistent with the effects of ASX from *Haematococcus pluvialis* [8,11,30]. 

ROS are associated with the formation of senile plaques in the brains of patients with AD, which may result in neuronal death [31,32]. Our results showed that scavenging of ROS by orally administered ASX from white shrimp shells significantly reduced neuronal loss in both cortex and hippocampus when compared to vehicle-AD group (*p* < 0.05) as shown in Figure 7.

We observed that amyloid plaque formation or aggregation via immunohistochemistry staining of amyloid beta peptide. The results shown in Figure 8, display that ASX significantly reduced positive staining of Aβ_1-42_ when compared to the vehicle-AD group (*p* < 0.05). The results were consistent with Rahman et al. (2019) [8] who reported that ASX treatment showed significant protection against Aβ mediated neuronal damage and forming plaque in the hippocampus.

In the last decade, the encapsulation technique has been shown to be an effective method to protect bioactive compounds within a shell and provides many other advantages. ASX is a highly unsaturated molecule and naturally not stable, and thus its functional ability is easily degraded during processing and storage. Another disadvantage is that ASX from shrimp shells may change the color and flavor of food. Encapsulation is an effective means to protect the stability of ASX within the selected wall material also it can help increase the surface area for release to the intestine [33]. Our results showed that AP could improve learning both spatial and non-spatial memory compared with AE, as shown in Figure 1 and Figure 2. In addition, AP showed significantly decreased oxidative damage and amyloid plaque level, and increased survival of neurons in both cortex and hippocampus areas when compared with those of the AE treatment groups. Therefore, use of encapsulation technology to improve the stability and efficacy of ASX extract in powder form is recommended for potential application in disease prevention of AD.

## 4. Materials and Methods 

### 4.1. Drugs and Chemicals 

ASX was extracted from fresh Pacific white shrimp (*Litopenaeus vannamei*) shells. Alginate was purchased from Union Chemical 1986 Co., Ltd. (Bangkok, Thailand). Modified starch was supplied by National Starch and Chemical (Bangkok, Thailand). Commercial ASX was purchased from GmbH (Staufen, Germany). Amyloid beta (1-42) peptides (Sigma, Camarillo, CA, USA), protein carbonate content assay kit (Sigma, USA), BCA protein assay kit (#23225, Thermo Scientific, Rockford, IL, USA), Anti-beta Amyloid antibody (ab2539, Abcam Inc, Cambridge, MA, USA) and Goat Anti-Rabbit IgG H&L (HRP) (ab6721, Abcam Inc, Cambridge, MA, USA) were also procured for the study. All chemicals and reagents used in this study were of AR grade.

### 4.2. Animals 

Adult male Wistar rats, eight weeks old, weighing 250–300 g were obtained from Nomura Siam International Co., Ltd. Rats were maintained (at room temperature: (22 ± 3 °C) with humidity of: 55% ± 10%), on a 12 h light-dark cycle (lights on 06.00–18.00), and with *ad libitum* access to water and foods. This study was conducted in accordance with the recommendations in the guide for the care and use of animal care outlined by the Faculty of Science, Prince of Songkla University (MOE 0521.11/105).

### 4.3. Extraction of ASX from Shrimp Shells and Preparation of Encapsulated ASX (ASX-powder) 

ASX extraction was prepared according to a previously described method [13]. To produce the ASX-powder, the emulsions were prepared by homogenizing the crude ASX (2 g/100 mL of wall material solution) with the wall material solution (alginate (2.0 g/100 mL) and modified starch (20 g/100 mL), mass ratio of 1:1) using a homogenizer (IKA T-25-Werke Gmbh & Co.KG, Staufen, Germany) at a rotational speed of 10,000 rpm for 20 min. The emulsion was converted into powders using cryogenic incorporated with freeze drying. The ASX-powder was collected and, stored in amber bottles.

### 4.4. Experimental Procedures 

The rats were randomly divided into eight groups, each group consisted of eight rats, as follows: Group 1 was the control group (C); group 2 served as a sham control group (SO) and received artificial cerebrospinal fluid (ACSF); groups 3–8 were Aβ_1-42_ infused groups in that they were injected intracerebroventricularly (i.c.v.) with amyloid beta peptides Aβ_1-42_ by using stereotaxic apparatus. To produce neurotoxicity in groups 3–8, 10 µL of normal saline-dissolved Aβ_1-42_ at a concentration of 2 mg/mL, was injected bilaterally on the first day. After injection of Aβ_1-42_, infused group 3 received propylene glycol (negative control; V), infused group 4 received ASX extract at 10 mg/kg/day p.o. (AE), infused group 5 received commercial ASX at 10 mg/kg/day p.o. (AC), infused group 6 received ASX powder at 10 mg/kg/day p.o. (AP), infused group 7 received powder blank at 10 mg/kg/day p.o. (BP), and infused group 8 received vitamin E at 100 mg/kg/day p.o. (VE). All treatments were orally administered for 30 days after Aβ_1-42_ infused. On days 14 and 29 post injection, animals were observed in the following behavioral tests: Morris water maze test, object location test, and object recognition test. The animals in all groups were assessed on spontaneous locomotor behaviors before doing all experiment.

### 4.5. Behavioral Studies 

#### 4.5.1. Morris Water Maze Test 

The Morris water maze test (MWM) was developed by Richard Morris in 1981 [34], and is a commonly used cognitive and behavioral assessment tool to evaluate both working and long-term spatial memory. Results of the test are thought to reflect the subject’s hippocampus function. MWM as shown in Appendix A, was performed on days 1–5 for training phases and on 14 and 29 day after oral administration compound of our experimental protocol, for testing phases. In the training phase, each rat was given four swimming trials per day for five consecutive days. In the testing phase, each rat receives the swimming trial or probe trial for the searching hidden platform, and the escape latency time was recoded. Twenty-four h later, the swimming trials or probe trial were conducted again, in which the platform was removed and rats were free to swim in the pool for 60 s [35]. Time spent by the rat in the target quadrant (target quadrant occupancy or located that hidden platform stand) for the search of the removed platform were recorded. Data were recorded by using video-image camera connected with a computer analysis system to observe swimming time.

#### 4.5.2. Object Location Test 

The object location memory test is used to assess cognition, particularly in spatial memory and discrimination, in rodent models of central nervous system disorders [36]. The fundamental concept of this test is that rodents are prone to give more time exploring a novel object than a familiar object and can remember when an object has been moved to a new location as shown in Appendix A. In our study, the test occurred in an open field arena, to which the animals were habituated beforehand. After habituation, two objects made of similar material were introduced to the arena. During the familiarization trial, the animals were allowed to explore the arena with the two objects for 3 min. After intervals of 30 min and 24 h, testing trials were done, in which one of the objects had been moved to a new location. Between trials, the objects and test areas were cleaned with 70% (v/v) ethanol to remove odor cues for later trials. The exploration activity, defined as sniffing while directing the nose toward and within 2 cm of the object, was separately scored for each object. Finally, the scores were used to calculate the discrimination index (ID) according to the formula: ID = (t [novel] − t [familiar])/ (t [novel] + t [familiar]) × 100.

#### 4.5.3. Object Recognition Test 

The object recognition test was used to assess the subject’s perception of color, shape, and texture as shown in Appendix A. In this test, each rat received two consecutive 3 min object exploration trials, separated by 1 and 24 h inter-trial intervals. During the familiarization trial, rats were individually presented with two similar objects. In the second trial, one of the two objects was randomly selected and replaced with a third, novel object. During the two trials, exploration of each object was defined as sniffing, licking, chewing, or having moving vibrissae while directing the nose toward the object, while within 2 cm of the object. Time of exploration was recorded separately for each object. Between trials, the objects and test areas were cleaned with 70% (v/v) ethanol to remove odor cues. Finally, the discrimination index was calculated as time spent exploring the novel object compared with the familiar object relative to the total time spent exploring all objects, according to the formula: Discrimination Index = (t [novel] − t [familiar])/(t [novel] + t [familiar]) × 100.

### 4.6. Preparation of Brain (Cortex and Hippocampus) Tissues Homogenate and Protein Extraction 

After completion of the experiment, the animals were sacrificed and the brains were quickly collected. The hippocampus and cortex (*n* = 5) were dissected and the tissues were frozen and stored at −80 °C. The cortex and hippocampus tissues were homogenized in 0.2 M phosphate buffer saline (PBS) followed by centrifugation at 13,000 rpm at 4 °C for 25 min. Supernatants were collected and stored at −80 °C for later determination of GPx activity and the levels of protein carbonyl, MDA, and superoxide anion.

#### 4.6.1. Measurement of Glutathione Peroxidase (GPx) Activity 

Glutathione peroxidase (GPx) activity was determined by previously published methods [37], in which the activity is measured indirectly by a coupled reaction with glutathione reductase. Oxidized glutathione, produced upon reduction of hydrogen peroxide by glutathione peroxidase, is recycled to its reduced state by glutathione reductase and NADPH. The oxidation of NADPH to NADP^+^ is accompanied by a decrease in absorbance at 340 nm. The rate of decrease in the A340 nm is directly proportional to the glutathione peroxidase activity. Our final 1 mL of system mixture contained 48 mM sodium phosphate, 0.38 mM EDTA, 0.12 mN β-NADPH, 0.95 mM sodium azide, 3.2 units of glutathione reductase, 1 mM glutathione (GSH), 0.02 mM DL-dithiothreitol, 0.0007% H_2_O_2_, and the standard enzyme glutathione peroxidase solution or a homogenate brain sample. The glutathione peroxidase solution was used as a standard for enzyme activity. The standard curve was plotted as the rate of A340 nm per min against GPx activity. One unit activity was defined as the amount of enzyme necessary to catalyze the oxidation by H_2_O_2_ of 1 µmole of GSH to GSSG per min at pH 7 at 25 °C. Data were reported as units of GPx per mg protein.

#### 4.6.2. Superoxide (O_2_^−^) Anion Assay 

The O_2_^−^ level was determined by spectrophotometric procedure according to Ukeda et al. [38], and based on a xanthine/xanthine oxidase (XO) assay which is marked by a color change from nitro blue tetrazolium (NBT)-yellow to formazan-blue. The reagent mixture included ethylenediaminetetraacetic acid (EDTA), NBT, xanthine, and xanthine oxidase (XO), along with sample. Absorbance of formazan chromophore was measured at 560 nm against blank and standard curves of TEMPOL. The data were expressed as % inhibition, as calculated following the Equation (1).
% inhibition= (A − B)/B × 100(1)

A = OD of reagent only and B = OD of reagent and sample.

#### 4.6.3. Measurement of Protein

Protein carbonyl, produced during protein oxidation in the biological sample, was measured by using a commercial assay kit (Sigma, USA). 

#### 4.6.4. Measurement of Malondialdehyde (MDA) 

Malondialdehyde (MDA) is produced during lipid peroxidation, and was measured using the methods of Ohkawa et al. [39]. A calibration curve was prepared using 1,1,3,3-tetramethoxypropane (TMP). A volume of 0.2 mL sample supernatant was mixed with 1.5 mL acetic acid (20%) at pH 3.5, 1.5 mL thiobarbituric acid (0.8%), and 0.2 mL sodium dodecyl sulphate (8.1%) and were added to processed tissue samples (0.1 mL), and then heated at 100 °C for 60 min, then cooled. A volume of 5 mL of n-butanol-pyridine (15:1) and 1 mL of distilled water were added and the mixture was vortexed vigorously. After centrifugation at 4000 rpm for 10 min, the organic layer was withdrawn and absorbance was measured at 532 nm using a spectrophotometer (ASYS UVM340, Biochrom Ltd., Cambridge, UK). Concentration of MDA was expressed as nmol/g tissue. The total brain protein assay was analyzed by the methods of Lowry et al. [40] with slight modification using Pierce BCA assay kit.

### 4.7. Histopathological Analysis of Brain 

#### 4.7.1. Cresyl Violet Staining for Nissl Substance 

After test animals were sacrificed, the whole brains (*n* = 3) were fixed in Bouin solution (Bio-optica, Milano, Italy) for 72 h. The brains were washed with 0.1 M phosphate buffered saline (PBS) and dehydrated sequentially in 70%, 80%, 90%, and 100% ethanol and then transferred to toluene. They were infiltrated and embedded in paraffin, sliced to a thickness of 5 μm using a rotary microtome, Leica RM2235 (Leica Microsystems, Nussloch, Germany), and then placed on silane-coated slides. They were deparaffinized with xylene, rehydrated in 100%, 95%, 90%, 80%, and 70% alcohol, and finally stained with cresyl violet (0.5%) to determine the neuronal density [41] in hippocampal CA1, CA3, and cortex regions. Neurons morphology was observed and captured by a Nikon E600 microscope with a DXM 1200 digital camera, using ACT-1 software (Nikon Inc., Badhoevedorp, The Netherlands). For all experimental groups, the surviving neurons were counted and recorded as number of neuron cells (size greater than 7 µm) using ImageJ (Java 1.4.2, U. S. National Institutes of Health, Bethesda, MA, USA).

#### 4.7.2. Immunohistochemistry Analysis

For immunohistochemistry, the brain sections from the different groups were deparaffinized with xylene and rehydrated in ethanol as above. Endogenous peroxidase was removed using 3% H_2_O_2_ in 100% methanol for 90 min. After washing three times with 0.1 M PBS containing Tween-20 (PBST), the sections were incubated in 1% glycine in 0.1 M PBST for 15 min and then washed three times with 0.1 M PBST. The brain sections were then incubated in blocking solution containing bovine serum albumin (4%) and 2% normal goat serum in 0.1 M PBST for 2 h. The slides were then incubated with primary antibody: Anti-beta amyloid antibody (1:200; catalog no. ab 2529, Abcam, CA, USA) at room temperature overnight. They were washed three times for 5 min each with PBST and were then incubated with secondary antibody: HRP-conjugated goat anti-rabbit IgG (1:1000; catalog no. ab 6721, Abcam) at room temperature for 2 h. After washing three times with PBST, liquid DAB-Plus Substrate Kit (Invitrogen, Camarillo, CA, USA) was added onto the brain sections to develop the color for 2 min and the reaction was then stopped by RO water. The slides were counter-stained with Mayer’s hematoxylin and mounted with permount. The results were recorded using a Nikon E600 microscope with a DXM 1200 digital camera, and ACT-1 software. Data were expressed as positive staining of β-amyloid (% area; total area covered with plaques relative to the total area) using ImageJ.

### 4.8. Statistical Analysis 

All data are presented as the mean ± standard error of mean (SEM). Significant differences between groups were analyzed using one-way ANOVA followed by the LSD’s post hoc test. Statistical significance was assumed at a *p* < 0.05. Statistical analysis was performed using the SPSS 17.0 software package for Windows.

## 5. Conclusions 

Here, we observed that after oral administration for one month, ASX from shrimp (*Litopenaeus vannamei*) shells protected Wistar rats from Aβ_1-42_ induced oxidative stress and resulted in improved learning and memory. Our study aimed to compare the effects of ASX in various forms, and we found that ASX powder showed better protection against cognitive dysfunction than the ASX extract at the same dose of compound. On the basis of these results, we suggest that ASX powder may be a promising candidate as a functional food product to protect from brain disorders, including in AD therapy.

## Figures and Tables

**Figure 1 marinedrugs-17-00628-f001:**
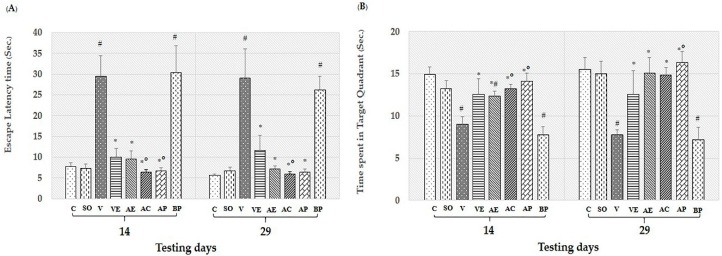
Effect of astaxanthin on Morris water maze test performance of rats. Escape latency time (**A**), time spent in target quadrant (**B**) in control rats (C), sham operate (SO): Artificial cerebrospinal fluid (ACSF), vehicle plus β-amyloid protein (Aβ)_1-42_ treated group (V), vitamin E: 100 mg/kg body weight (BW) plus Aβ_1-42_ treated group (VE), astaxanthin extract: 10 mg/kg BW plus Aβ_1-42_ treated group (AE), commercial astaxanthin: 10 mg/kg BW plus Aβ_1-42_ treated group (AC), astaxanthin powder: 10 mg/kg BW plus Aβ_1-42_ treated group (AP), and blank powder: 10 mg/kg BW plus Aβ_1-42_ treated group (BP). Values are expressed as mean ± SEM (*n* = 8). # -*p* < 0.05 compared with control group; * -*p* < 0.05 compared with vehicle and blank powder groups; °- *p* < 0.05 compared with astaxanthin extract group.

**Figure 2 marinedrugs-17-00628-f002:**
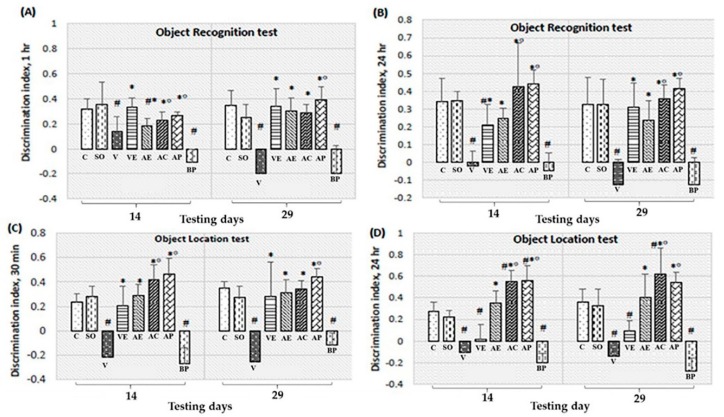
Effect of astaxanthin on discrimination index after 1 (**A**) and 24 h (**B**) of rats presented with object recognition test and effect of astaxanthin on discrimination index after 30 min (**C**) and 24 h (**D**) of rats presented with object location test. Control rats (C), sham operate (SO): Artificial cerebrospinal fluid (ACSF), vehicle plus Aβ_1-42_ treated group (V), vitamin E: 100 mg/kg BW plus Aβ_1-42_ treated group (VE), astaxanthin extract: 10 mg/kg BW plus Aβ_1-42_ treated group (AE), commercial astaxanthin: 10 mg/kg BW plus Aβ_1-42_ treated group (AC), astaxanthin powder: 10 mg/kg BW plus Aβ_1-42_ treated group (AP), and blank powder: 10 mg/kg BW plus Aβ_1-42_ treated group (BP). Values are expressed as mean ± SEM (*n* = 8). # -*p* < 0.05 compared with control group; * -*p* < 0.05 compared with vehicle and blank powder groups; °- *p* < 0.05 compared with astaxanthin extract group.

**Figure 3 marinedrugs-17-00628-f003:**
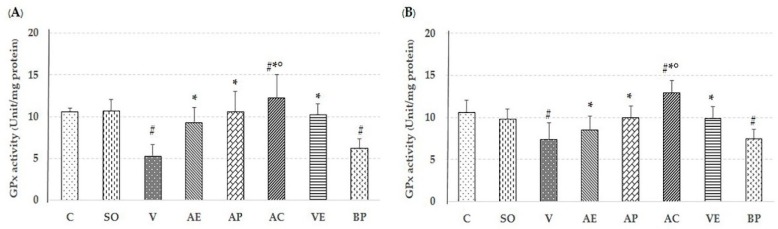
Effect of astaxanthin on glutathaione peroxidase (GPx) enzyme activity in hippocampus area (**A**) and cerebral cortex area (**B**). Control rats (C), sham operate (SO): ACSF, vehicle plus Aβ_1-42_ treated group (V), vitamin E: 100 mg/kg BW plus Aβ_1-42_ treated group (VE), astaxanthin extract: 10 mg/kg BW plus Aβ_1-42_ treated group (AE), commercial astaxanthin: 10 mg/kg BW plus Aβ_1-42_ treated group (AC), astaxanthin powder: 10 mg/kg BW plus Aβ_1-42_ treated group (AP), and blank powder: 10 mg/kg BW plus Aβ_1-42_ treated group (BP). Values are expressed as mean ± SEM (*n* = 5). # -*p* < 0.05 compared with control group; * -*p* < 0.05 compared with vehicle and blank powder groups; °- *p* < 0.05 compared with astaxanthin extract group.

**Figure 4 marinedrugs-17-00628-f004:**
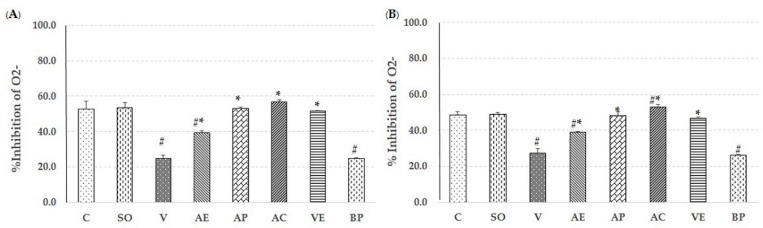
Effect of astaxanthin on % inhibition of o^2-^ in hippocampus area (**A**) and cerebral cortex area (**B**). Control rats (C), sham operate (SO): ACSF, vehicle plus Aβ_1-42_ treated group (V), vitamin E: 100 mg/kg BW plus Aβ_1-42_ treated group (VE), astaxanthin extract: 10 mg/kg BW plus Aβ_1-42_ treated group (AE), commercial astaxanthin: 10 mg/kg BW plus Aβ_1-42_ treated group (AC), astaxanthin powder: 10 mg/kg BW plus Aβ_1-42_ treated group (AP), and blank powder: 10 mg/kg BW plus Aβ_1-42_ treated group (BP). Values are expressed as mean ± SEM (*n* = 5). # -*p* < 0.05 compared with control group; * -*p* < 0.05 compared with vehicle and blank powder groups; °- *p* < 0.05 compared with astaxanthin extract group.

**Figure 5 marinedrugs-17-00628-f005:**
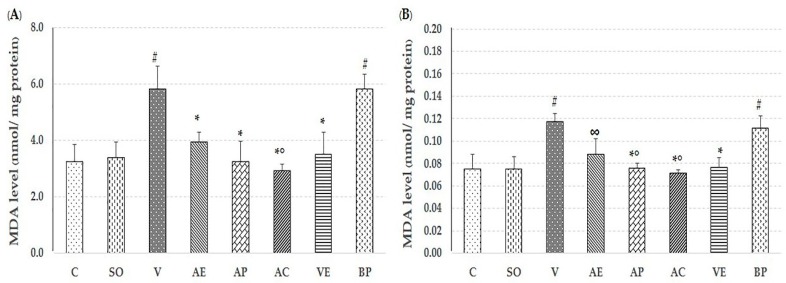
Effect of astaxanthin on malondialdehyde (MDA) level in hippocampus area (**A**) and cerebral cortex area (**B**). Control rats (C), sham operate (SO): ACSF, vehicle plus Aβ_1-42_ treated group (V), vitamin E: 100 mg/kg BW plus Aβ_1-42_ treated group (VE), astaxanthin extract: 10 mg/kg BW plus Aβ_1-42_ treated group (AE), commercial astaxanthin: 10 mg/kg BW plus Aβ_1-42_ treated group (AC), astaxanthin powder: 10 mg/kg BW plus Aβ_1-42_ treated group (AP), and blank powder: 10 mg/kg BW plus Aβ_1-42_ treated group (BP). Values are expressed as mean ± SEM (*n* = 5). # -*p* < 0.05 compared with control group; * -*p* < 0.05 compared with vehicle and blank powder groups; °- *p* < 0.05 compared with astaxanthin extract group; ∞ -*p* < 0.05 compared with vehicle group.

**Figure 6 marinedrugs-17-00628-f006:**
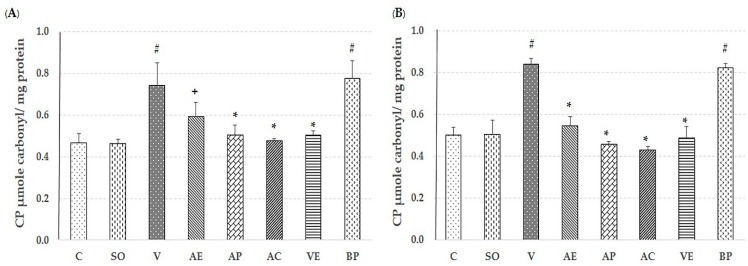
Effect of astaxanthin on protein carbonyl level in hippocampus area (**A**) and cerebral cortex area (**B**). Control rats (C), sham operate (SO): ACSF, vehicle plus Aβ_1-42_ treated group (V), vitamin E: 100 mg/kg BW plus Aβ_1-42_ treated group (VE), astaxanthin extract: 10 mg/kg BW plus Aβ_1-42_ treated group (AE), commercial astaxanthin: 10 mg/kg BW plus Aβ_1-42_ treated group (AC), astaxanthin powder: 10 mg/kg BW plus Aβ_1-42_ treated group (AP), and blank powder: 10 mg/kg BW plus Aβ_1-42_ treated group (BP). Values are expressed as mean ± SEM (*n* = 5). # -*p* < 0.05 compared with control group; * -*p* < 0.05 compared with vehicle and blank powder groups; + -*p* < 0.05 compared with blank powder group.

**Figure 7 marinedrugs-17-00628-f007:**
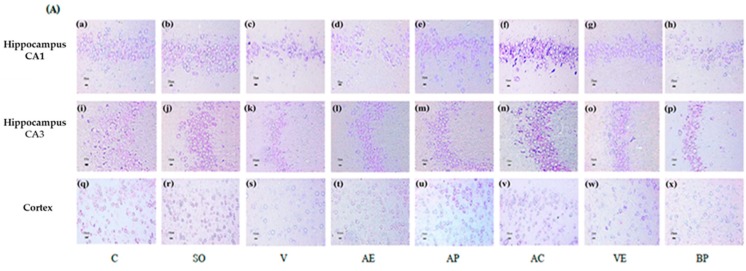
Neuroprotective effect of astaxanthin. (**A**) Cresyl violet staining was performed on sections from the hippocampus and cortex to inspect the survival neurons in the hippocampus CA1 (**a**–**h**), CA3 (**i**–**p**), and cortex (**q**–**x**) region (scale bar = 25 µm). Number of neuron cells in CA1 (**B**), CA3 (**C**), and cortex region (**D**). Control rats (C), sham operate (SO): ACSF, vehicle plus Aβ_1-42_ treated group (V), vitamin E: 100 mg/kg BW plus Aβ_1-42_ treated group (VE), astaxanthin extract: 10 mg/kg BW plus Aβ_1-42_ treated group (AE), commercial astaxanthin: 10 mg/kg BW plus Aβ_1-42_ treated group (AC), astaxanthin powder: 10 mg/kg BW plus Aβ_1-42_ treated group (AP), and blank powder: 10 mg/kg BW plus Aβ_1-42_ treated group (BP). Values are expressed as mean ± SEM (*n* = 3). # -*p* < 0.05 compared with control group; * -*p* < 0.05 compared with vehicle and blank powder groups; °- *p* < 0.05 compared with astaxanthin extract group; + -*p* < 0.05 compared with vitamin E.

**Figure 8 marinedrugs-17-00628-f008:**
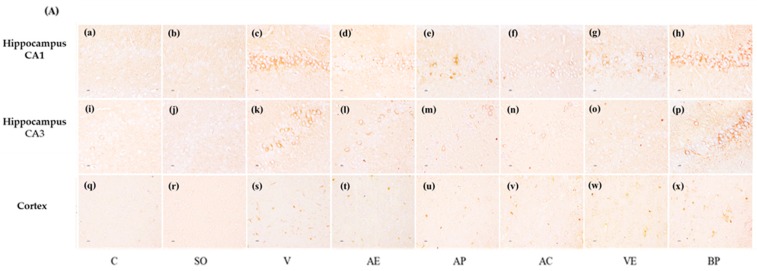
Neuroprotective effect of astaxanthin. (**A**) Immunohistochemistry of sections from the hippocampus CA1 (**a**–**h**), CA3 (**i**–**p**), and cortex (**q**–**x**) region. Positive staining of β-amyloid (% area) in CA1 (**B**), CA3 (**C**), and cortex (**D**) region (scale bar = 25 µm). Control rats (C), sham operate (SO): ACSF, vehicle plus Aβ_1-42_ treated group (V), vitamin E: 100 mg/kg BW plus Aβ_1-42_ treated group (VE), astaxanthin extract: 10 mg/kg BW plus Aβ_1-42_ treated group (AE), commercial astaxanthin: 10 mg/kg BW plus Aβ_1-42_ treated group (AC), astaxanthin powder: 10 mg/kg BW plus Aβ_1-42_ treated group (AP), and blank powder: 10 mg/kg BW plus Aβ_1-42_ treated group (BP). Values are expressed as mean ± SEM (*n* = 3). # -*p* < 0.05 compared with control group; * -*p* < 0.05 compared with vehicle and blank powder groups; °- *p* < 0.05 compared with astaxanthin extract group; + -*p* < 0.05 compared with vitamin E.

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
