# Peer review of "Effects of Astaxanthin from Shrimp Shell on Oxidative Stress and Behavior in Animal Model of Alzheimer’s Disease"

_marinedrugs, 2019, doi:10.3390/md17110628_

Round 1
Reviewer 1 Report
Abstract should be worked on to give the reader a better sense of all that is being examined. Eg testing the different AXT and formulations.
English needs a little work through the paper. Eg. In the abstract it says “biochemical histological analysis” this may mean Biochemical and histological
figure one would be better if labels on the x axis instead of just the legend. It is hard to read the legends esp if the bar is small. This is true for other figures too, eg labels on x axis for figure 7 graphs
morris water maze data: states that “Time spent in the target quadrant (target quadrant occupancy) and number of times that the animal crossed the position where the platform was placed (crossing times) were recorded.” But this is not clear from Figure 1 which shows escape latency and retention times. It is unclear what they mean by retention time? and escape latency is time to platform but they are doing a probe trial with no platform.
What is swimming trial? Is this with a viable platform?
water maze task is 5 days long just for the testing, but state tested on day 29 and euthanized on day 31. How is that possible?
Were the same animals used in all tasks? Were the same animals tested on day 14 and 29? Repeating learning and memory tasks is not a good practice
figure 8 is not measuring plaques but immune positive A-beta as percentage area. Label needs changing
Orally administered? Does this mean Gavage or in Diet?
Unclear about brain preparation. 4.7.1 and 4.6. It is very unclear how many animals are used here. There are two different methods of brain extraction but no mention of different cohorts.
What stereological software was used for cell counts?
Reviewer 2 Report
Taksima et al described the usefulness of astaxanthin by using rat AD model. The positive effects in the water-maze test were excellent. However, the results of histological investigation were unclear.
Abeta plaque was reduced by astaxanthin treatment in the figure 8. However, representative image did not clearly show typical Abeta plaque. The beneficial effects of astaxanthin on Abeta should be verified by another experiment. ELISA would be good.
Oxidative damage and inflammation are important features of the brain pathology of AD. The effects of astaxanthin on neuro-inflammation should be examined as well.
In the Morris water maze test, motor function may affect the results. Not only latency time and retention time, but also total distance or swimming speed should be presented. If motor function was not evaluated in the Morris water maze test, another test should be considered.
Round 2
Reviewer 1 Report
Authors seemed to address previous concerns
Author Response
Thank you so much for your valuable comments for this manuscript.
Reviewer 2 Report
The manuscript has been much improved and is in a nice condition now.
Author Response

(The authors gave the same response as above.)
